# The Response of *Paracoccidioides lutzii* to the Interaction with Human Neutrophils

**DOI:** 10.3390/jof9111088

**Published:** 2023-11-07

**Authors:** Lana O’Hara Souza Silva, Lilian Cristiane Baeza, Laurine Lacerda Pigosso, Kleber Santiago Freitas e Silva, Maristela Pereira, Marcos Antonio Batista de Carvalho Júnior, Célia Maria de Almeida Soares

**Affiliations:** 1Laboratório de Biologia Molecular, Instituto de Ciências Biológicas, Universidade Federal de Goiás, Goiania 74690-900, GO, Brazil; lanaohara.loss@gmail.com (L.O.S.S.); laurinepigosso@gmail.com (L.L.P.); smallbinho@hotmail.com (K.S.F.e.S.); maristelaufg@gmail.com (M.P.); carvalhojr.biolic@gmail.com (M.A.B.d.C.J.); 2Laboratório de Bacteriologia e Micologia Médica, Centro de Ciências Médicas e Farmacêuticas, Universidade Estadual do Oeste do Paraná, Cascavel 85819-110, PR, Brazil; lilianbaeza@gmail.com

**Keywords:** genus *Paracoccidioides*, human neutrophils, proteomics, metabolism shunt, stress response

## Abstract

The fungal pathogen *Paracoccidioides lutzii* causes systemic mycosis Paracoccidioidomycosis (PCM), which presents a broad distribution in Latin America. Upon infection, the fungus undergoes a morphological transition to yeast cells and provokes an inflammatory granulomatous reaction with a high number of neutrophils in the lungs. In this work, we employed proteomic analysis to investigate the in vitro response of the fungus to the interaction with human neutrophils. Proteomic profiling of *P. lutzii* yeast cells harvested at 2 and 4 h post interaction with human polymorphonuclear cells allowed the identification of 505 proteins differentially accumulated. The data indicated that *P. lutzii* yeast cells underwent a shift in metabolism from glycolysis to Beta oxidation, increasing enzymes of the glyoxylate cycle and upregulating enzymes related to the detoxification of oxidative and heat shock stress. To our knowledge, this is the first study employing proteomic analysis in the investigation of the response of a member of the *Paracoccidioides* genus to the interaction with neutrophils.

## 1. Introduction

Paracoccidioidomycosis (PCM) is caused by the thermally dimorphic fungi of the genus *Paracoccidioides*. At present, PCM is spread in Central and South America, with a higher prevalence in Brazil [1]. Up to now, seven species of this genus have been identified, and *Paracoccidioides lutzii* predominates in Brazil, in the Midwest [2]. Members of the genus grow as mold in the environment but undergo morphotype conversion to the yeast phase when infecting human tissues [3]. The disease is characterized by a chronic inflammatory granulomatous reaction as a consequence of a Th-1-mediated adaptive immune response [4]. The immune response leads to the accumulation of neutrophils and micro abscess formation, and with lesion progression, macrophages replace neutrophils. The high number of neutrophils found in *Paracoccidioides* spp. granulomas demonstrates the importance of polymorphonuclear neutrophil (PMN) cells during this infection [5].

Neutrophils are efficient phagocytic cells, as demonstrated in experimental infections such as those provoked by Listeria [6], Legionella pneumophila [7], Mycobacterium [8], Candida albicans [9], Toxoplasma gondii [10] and Trypanosoma cruzi [11]. At the site of any inflammatory insult, neutrophils are recruited, and their activation initiates oxidative burst, leading to the production of reactive oxygen species (ROS) [12]. Neutrophils are the first line of response at the site of infection; they are capable of recognizing and phagocytizing pathogens. They provoke microbial destruction through oxidative and non-oxidative mechanisms by using toxic reactive oxygen enzymes and antimicrobial peptides and through extracellular traps composed of its genetic material [13].

Neutrophils express receptors on their surface, including TLR2, TLR4 and dectin-1, which mediate fungal cell recognition, while other receptors, such as FCγR and CR3, help opsonization and later phagocytosis [14]. These cells can phagocytize microorganisms, incorporating them into their phagolysosomes and destroying them intracellularly. They produce large amounts of toxic oxygen-derived products, such as hydrogen peroxide (H_2_O_2_) and superoxide anion (O^−2^), via the NADPH oxidase system [15]. Degranulation is important for neutrophils to exercise their microbicidal capacity. Between the components found in the granules of neutrophils, we have myeloperoxidase, an enzyme that is involved in the synthesis of hypochlorous acid, and lactoferrin, which is involved in the uptake of iron. The production of extracellular neutrophil traps (NETs), composed of chromatin and microbicidal proteins, such as elastase, cathepsin, and calprotectin, contributes to the microbicidal action [16]. The formation of NETs against conidia and yeasts of *P. brasiliensis* has been demonstrated; however, these structures seem to exert a fungistatic effect, demonstrating that *P. brasiliensis* could evade the action of NETs [17]. Intracellularly, *P. brasiliensis* can induce an anti-apoptotic process to create an environment for its survival and replication [18]. By using electron microscopy and cytochemical approaches, authors have indicated the participation of human neutrophils in killing *P. brasiliensis* [19]. Accordingly, neutrophils can kill ingested and extracellular fungi. Cytochemical studies have shown that once *P. brasiliensis* attaches to the neutrophil surface, it triggers a respiratory burst with the release of oxygen-derived products. Attachment also triggers neutrophils’ degranulation, with the release of endogenous peroxidase localized in cytoplasmic granules. Together, these processes could lead to the killing of both ingested and extracellular *P. brasiliensis*.

In this work, we evaluated the proteome of a species of the *Paracoccidioides* genus, *P*. *lutzii*, which predominates in the middle west of Brazil. Our studies evidenced a rearrangement of the fungus metabolism, including the induction of the glyoxylate cycle and the response to oxidative stress when *P. lutzii* is associated with neutrophils. On the other hand, an initial analysis of the neutrophil proteome reveals that those cells express proteins related to the host’s response to the pathogen. To our knowledge, this is the first study demonstrating proteins expressed by the fungus and by neutrophils during *P. lutzii* interaction.

## 2. Materials and Methods

### 2.1. Microorganisms and Culture Conditions

*P. lutzii* (ATCC MYA-826) was grown in Fava Netto’s solid medium [20] at 36 °C for 7 days. Prior to this culture, the fungus was periodically recovered from interactions with J774 1.6 murine macrophages to ensure the maintenance of virulence. For the experiments, yeast cells were inoculated in Fava Netto’s liquid medium at 35 °C under constant agitation (150 rpm) for 48 h to conduct the experiments, as detailed below.

### 2.2. Isolation of Polymorphonuclear Cells from Human Peripheral Blood

Peripheral blood from healthy donors (age range 20–40 years) was collected via venipuncture. The volunteers signed a consent form; this study was approved by the Committee for Ethics in Research Involving Humans at the Universidade Federal de Goiás (UFG)/Goiânia, Goiás, Brazil (No. 019/13). PMNs were separated via density gradient centrifugation using the Histopaque 1077 (Sigma- Aldrich, St. Louis, MO, USA) at 500× *g* for 30 min, followed by erythrocytes lysis with a hypotonic solution (NaCl 0.2%) [21]. Cellular viability was assessed through the use of a trypan blue dye exclusion test and then determined in a Neubauer chamber (Olen, Kasvi, São José dos Pinhais, Brazil). Purified PMNs (95% of the cells) were then resuspended in RPMI medium 1640 (Vitrocell. Rio de Janeiro, Brazil). The cell culture was adjusted using a Neubauer chamber (Olen, Kasvi, São José dos Pinhais, Brazil) for 2.5 × 10^6^ cells/mL before all procedures.

### 2.3. Internalization Assay

The experimental workflow is summarized in Appendix A. For the internalization assay, the yeast cells obtained from a 48 h inoculum in Fava Netto’s liquid medium were passed through a sterilized gauze with the aid of a syringe and needle. The cellular density was adjusted using a Neubauer chamber (Olen, Kasvi, São José dos Pinhais, Brazil) to 5 × 10^6^ yeasts/mL in phosphate-buffered saline (PBS). Prior to incubation with PMNs, the yeast cells were opsonized with 2.5% serum (*v*/*v*) from the healthy donors for 30 min at room temperature. The interaction was carried out in a proportion of yeast cells: PMNs of 2:1, followed by incubation for 2 and 4 h at 36 °C and 5% CO_2_ in RPMI medium. The control was obtained by incubating 5 × 10^6^ opsonized yeast cells per well in RPMI medium for 2 and 4 h at 36 °C and 5% CO_2_.

### 2.4. Preparation of Protein Extract

Considering the proteomic analyses, seven independent internalization assays were performed, mixed and used to obtain the protein extracts. In summary, to obtain the three replicates, a total of 21 assays were performed. The fungus–neutrophil interaction was carried out in polystyrene plates with 12 wells; the volume used in each well was 2 mL. After infection periods of 2 and 4 h, part of the supernatant (about 1 mL) was slowly removed, and 800 µL of ice-cold ultrapure sterile water was added to lyse the PMN, followed by centrifugation at 10,000× *g* and 4 °C for 5 min. The material was washed once with RapiGEST SF Surfactant 0.2% (*v*/*v*) (Waters Corp., Milford, MA, USA), followed by 3 washes with ultrapure water and PBS 1X [22]. The pellet was resuspended in extraction buffer (20 mM Tris–HCl, pH 8.8, and 2 mM CaCl_2_) and distributed in tubes containing glass beads (Sigma–Aldrich, St. Louis, MO, USA) in equal volume of the material. The suspension was processed using BeadBeater equipment (BioSpec, Products Inc., Bartlesville, OK, USA) for 3 cycles of 30 s, with intervals of 1 min on ice. The cell lysate was centrifuged at 10,000× *g* for 15 min at 4 °C, followed by repeated centrifugation until no pellet formation was observed. Control cells were obtained by incubating *P. lutzii* yeast cells in RPMI medium, with the same washing procedures as for the internalization condition. Protein content in the supernatant was quantified using the Bradford assay using bovine serum albumin (Roche Diagnostics, Mannheim, Germany) as standard [23].

### 2.5. Digestion of Proteins Extracts for Mass Spectrometry

The proteins were enzymatically digested based on Murad et al. (2011) [24]. Then, 300 µg of protein from each time was added to 10 µL of 50 mM ammonium bicarbonate buffer, pH 8.5, in a microcentrifuge tube, and 150 µL RapiGEST^TM^ SF (0.2% *v*/*v*) (Waters, Milford, MA, USA) was added, vortexed and incubated for 15 min at 80 °C. Disulfide bonds were reduced via incubation with 2.5 µL of 100 mM dithiothreitol (DTT) (GE Healthcare, Piscataway, NJ, USA) at 60 °C for 30 min, and then alkylation was performed via incubation with 300 mM iodoacetamide (GE Healthcare, Piscataway, NJ, USA) at room temperature in the dark for 30 min. The proteins were digested overnight at 37 °C with 30 µL of Trypsin solution at 0.05 µg/µL (Promega, Madison, WI, USA). Digestion was stopped with the addition of 30 µL of 5% trifluoroacetic acid (*v*/*v*), followed by incubation at 37 °C for 90 min. The samples were centrifuged, and supernatants were dried in a speed vacuum. The peptides were suspended in a solution containing 20 mM ammonium formate and 150 fmol/µL of Rabbit Phosphorylase B (PHB; Waters, Manchester, UK) as an internal standard and transferred to a Waters Total Recovery vial (Waters Corp., Milford, MA, USA). The samples were placed in an auto-sampler and stored at 4 °C for chromatography and mass spectrometry analysis.

### 2.6. Chromatography and Mass Spectrometry

Chromatography was performed on a precolumn nanoEase^TM^ 5 mm × Bridge^TM^ BEH130 C18 300 mm × 50 mm; trap column 5 mm, 180 mm × 20 mm and analytical reversed-phase column BEH130 C18 1.7 mm, 100 mm × 100 mm (Waters Corp., Milford, MA, USA) using a nanoACQUITY UPLC^TM^ system (Waters Corp., Milford, MA, USA). The peptides were separated into 10 fractions, with a linear gradient of acetonitrile/0.1% (*v*/*v*) formic acid, as follows: 8.7, 11.4, 13.2, 14.7, 16, 17.4, 18.9, 20.7, 23.4 and 65%. The lock mass GFP [Glu]1-Fibrinopeptide B human ([M + 2H]^2+^ = 785.8426) at 200 fmol/µL (Sigma–Aldrich, St. Louis, MO, USA) was used to ensure accuracy during the analysis. Mas spectrometric experiments were performed on a Synapt G1 MS^TM^ (Waters, Manchester, UK) equipped with a NanoElectronSpray source and mass analyzers hybrid quadrupole/time-of-flight (TOF) operating in V-mode. The mass spectrometric full-scan data were acquired in the positive ion mode nano-ESI(+) in the *m*/*z* range of 50–2000 with a scan time of 0.4 s. The spectrometer was automatically programmed to switch between low collision energy MS (6 V) and elevated collision energies (40 V). The peptide fractions were analyzed in triplicates.

### 2.7. Raw Data Processing and Protein Identification Analysis

Raw files were analyzed together using the ProteinLynx Global Server software version 3.0.2 (PLGS) (Waters, Manchester, UK). The protein identifications and quantitative packaging were generated using specific algorithms [25,26], and the search was performed against a *P. lutzii* database (Broad Institute; http://www.broadinstitute.org/annotation/genome/paracoccidioides_brasiliensis/MultiHome.html, accessed on 3 April 2017). The parameters were as follows: strict trypsin specificity, allowing one missed cleavage; minimum peptide length was five amino acids; the detection of at least two fragment ions per peptide; carbamidomethylation of cysteine was a fixed modification; maximum protein mass (600 kDa); and phosphorylation STY of proteins and oxidation of methionine were set as variable modifications. A minimum of two peptides were required for protein identification, and peptide spectral matches were filtered using a target–decoy approach at a false discovery rate (FDR) of 4%. Proteins identified with low accuracy were excluded, and proteins with two or more peptides were admitted in at least two out of three technical replicate injections. The identified proteins were organized by the expression algorithm into a statistically significant list corresponding to induced and reduced regulation ratios between internalized and control. The mathematic model used to calculate the ratios is part of the expression algorithm inside the PLGS from Waters Corporation [27]. The software shows the expression analysis statistics as the induced proteins with a probability of upregulation of 0.95 or more and the reduced proteins with a probability of 0.05 or less. PLGS uses the following strategy for peptide identification: first, only completely cleaved tryptic peptides are used for identification (PepFrag1). The second pass of the database algorithm (PepFrag2) is designed to identify peptide modifications and non-specific cleavage products to proteins that were positively identified in the first pass. For the analysis of protein quantification levels, the observed intensity measurements were normalized with a protein that showed a variance coefficient and that was detected in all replicates. A 1.2%-fold change was used as a cutoff to determine the differentially abundant proteins in the internalized and control. Besides the 1.2%-fold change, another determination to define increased and decreased proteins was regulation at 4 h interaction time. Proteins that decreased at 2 h and increased at 4 h were considered increased and proteins that increased at 2 h and decreased at 4 h were considered decreased. Also, proteins that did not appear at 2 h but showed regulation at 4 h, even with fold change inferior to 0.83, were considered increased. The mass spectrometry proteomics data have been deposited in the PeptideAtlas repository (www.PeptideAtlas.org, accessed on 15 February 2023) with the data set identifier: PASS03809 (http://www.peptideatlas.org/PASS/PASS03809, accessed on 15 February 2023). Regulated proteins were functionally categorized using FungiDB (https://fungidb.org/fungidb/, accessed on 26 September 2022), Uniprot (https://www.uniprot.org, accessed on 28 September 2022) and KEGG (https://www.genome.jp/kegg/pathway.html, accessed on 28 September 2022) databases. The annotation of non-characterized proteins was performed via homology from the proteins present in the NCBI (https://www.ncbi.nlm.nih.gov/, accessed on 30 September 2022). For the experiment dynamic range, the peptide parts per million error (ppm) and peptide detection type using software such as MassPivot v1.0.1. and Spotfire v8.0. Microsoft Excel (Microsoft^®^), which was also used for table manipulations. The data obtained via mass spectrometry of the internalization assay was also analyzed against the Human neutrophil database (https://www.uniprot.org/uniprotkb?query=human%20neutrophil, accessed on 28 September 2022).

### 2.8. Statistical Analyses and Graphics Construction

Excel software (Microsoft Office 365) was used to run the statistical analysis, and data with *p*-values < 0.05 were considered statistically significant. Bar charts and heat maps were generated through the use of GraphPad Prism 8.0.1 software (San Diego, CA, USA).

## 3. Results

### 3.1. Proteomic Analysis of Yeast Cells Interacting with Neutrophils

In this study, we performed the proteomic profile of *P. lutzii* during the infective process in human neutrophils in 2 and 4 h. Microscopical analysis depicts *P. lutzii* internalized by human neutrophils, as demonstrated via staining the preparation with Giemsa (Figure 1).

The proteome allowed the identification of 591 proteins that were classified according to the functional category (Appendix A). Proteomic and bioinformatic analysis resulted in 505 differentially expressed proteins. A 1.2-fold change was used as a threshold in order to determine increased and decreased proteins. Among those, with a 1.2-fold change, 189 proteins increased (Appendix A), while 316 decreased (Appendix A) upon *P. lutzii* neutrophil interaction, as can be seen in Figure 2. Around 85% of the identified proteins were classified into biological categories; conversely, around 15% of them were uncategorized and could not be re-annotated via the Uniprot system.

### 3.2. Interaction with Human Neutrophils Promotes Increase in Proteins of the Oxidative Stress Response in P. lutzii

Stress response and detoxification were cellular processes induced in *P. lutzii* upon interaction with human neutrophils, as depicted in Table 1. Among the induced proteins in yeast cells interacting with human neutrophils, we can highlight those involved in the response to stress, such as the heat shock proteins; the expression of HSP90 (PAAG_05679), as an example, increased more than eight times in yeast cells upon 4 h of neutrophil interaction. Detoxification enzymes such as peroxisomal catalase (PAAG_01454), three superoxide dismutases, were induced, Cu/Zn-containing SOD1 (PAAG_04164), Fe/Mn-containing SOD2 (PAAG_02725) and Fe/Mn-containing SOD5 (PAAG_02926). Table 1 presents the cited enzymes, whose expression is depicted in Figure 3. This figure summarizes data from the proteomic analysis and provides insights into the mechanisms employed by this fungus in response to the oxidative stress of neutrophils. Superoxide dismutases mediate the detoxification of superoxide into hydrogen peroxide. The degradation of hydrogen peroxide is catalyzed by catalase. Of relevance, most of the proteins depicted in Table 1 increased over the time of *P. lutzii* exposure to neutrophils (Table 1 and Figure 3).

### 3.3. The Interaction of P. lutzii with Neutrophils Promotes Increase in Proteins Related to Acetyl-CoA Formation

Enzymes of beta oxidation were increased in yeast cells internalized by the neutrophils, as depicted in Table 2 and Figure 4. Carnitine O acetyltransferase (PAAG_06224) participates in the internalization of acyl-CoA into the mitochondrial matrix by conjugating acyl-CoA to carnitine. This protein was not found at 2 h but showed a small regulation after 4 h of neutrophil internalization, presenting an increase in expression over time. The peroxisomal multifunctional dimeric protein (MEF; PAAG_08859) contains enoyl-CoA hydratase-2 domain and a second domain that displays (S)-3-hydroxyacyl-CoA dehydrogenase [28]. A search for SKL domains—a serine–lysine–leucine consensus sequence of peroxisomal targeting—identified that *P. lutzii* MEF contains this consensus sequence, suggesting a possible activity of beta oxidation in the peroxisome of *P. lutzii*. Other proteins related to fatty acids oxidation, such as 3 hydroxybutyryl CoA dehydrogenase (PAAG_06329), enoyl CoA hydratase (PAAG_06309) and 3 ketoacyl CoA thiolase (PAAG_02664), were also increased after neutrophil internalization, suggesting a potential utilization of fatty acids as fuel within neutrophils.

The interaction with neutrophils also increased the expression of proteins related to amino acid catabolism. One subunit from 2-oxoisovalerate dehydrogenase (PAAG_01310), which generates acetyl-CoA through the degradation of valine, leucine and isoleucine, was increased after 4 h of neutrophil internalization. The glutaryl CoA dehydrogenase (PAAG_05984) transforms glutaryl-CoA into 2-trans-enoyl-CoA, which enters the beta oxidation or into crotonoyl-CoA in the lysine and tryptophan oxidation pathways, whose final product is acetyl-CoA. The enzymes from the pyruvate dehydrogenase complex, dihydrolipoyl dehydrogenase (PAAG_03330), pyruvate dehydrogenase E1 component (subunit alpha, PAAG_08295; subunit beta, PAAG_01534) and pyruvate dehydrogenase protein X component (PAAG_00050), were increased upon 4 h of neutrophil internalization. This complex is responsible for the oxidation and decarboxylation of pyruvate to produce acetyl-CoA. Dihydrolipoyl dehydrogenase can also participate in the degradation of valine, leucine and isoleucine, producing acetyl-CoA, together with 2-oxoisovalerate dehydrogenase and pyruvate dehydrogenase X component (PAAG_02769), which was exclusively found at both neutrophil interaction times. Most of the cited enzymes had their expression increased over time, suggesting a need for acetyl-CoA production during neutrophil internalization.

### 3.4. The Interaction of P. lutzii with Neutrophils Promotes Increase in Enzymes of TCA and Glyoxylate Pathway and Its Intermediates Formation

In the present assay, enzymes related to the TCA pathway increased upon neutrophil interaction, as depicted in Table 3 and Figure 5. One of those enzymes is 3-isopropylmalate dehydratase (PAAG_03296), an aconitase homologue, which catalyzes the reversible reaction of citrate into isocitrate. The enzymes ATP citrate synthase subunit 1 (PAAG_05150) and citrate synthase (PAAG_08075) catalyze the conversion of acetyl-CoA and oxaloacetate into citrate, the first step of the TCA pathway, increased after 4 h of neutrophil internalization. Both subunits of succinyl CoA ligase (PAAG_00417, PAAG_01463), which converts succinyl CoA into succinate, were increased upon neutrophil internalization. Our proteomic results also evidenced the increase in glutamate dehydrogenase (PAAG_01002). This enzyme catalyzes the conversion of L-glutamate to α-ketoglutarate and NH_3_ in the presence of NAD^+^, thus providing the compound to the TCA cycle. The enzyme alanine glyoxylate aminotransferase (PAAG_03138), which releases pyruvate, also increased. The enzymes 4 aminobutyrate aminotransferase (PAAG_00468) and succinate semialdehyde dehydrogenase (PAAG_08718) belong to the GABA shunt, where the first converts GABA and α-ketoglutarate into succinate semialdehyde, which is then converted into succinate by the second enzyme. The argininosuccinate lyase (PAAG_06407) releases arginine and fumarate, a TCA intermediate, also increased. Another increased enzyme was phosphoglycerate kinase (PAAG_02869), which produces 3-phosphoglycerate, an isocitrate dehydrogenase regulator, which leads to an increase in TCA [29]. The increase in malate synthase (PAAG_04542) and isocitrate lyase (PAGG_06951) suggests a possible heightening of the glyoxylate cycle. Enzymes of the beta oxidation (Table 2 and Figure 4) may potentially supply acetyl CoA to feed the TCA and the glyoxylate cycle. Figure 5 depicts the heat map considering the cited metabolic processes of energy metabolism.

### 3.5. An Overview of Metabolic Changes in P. lutzii Interacting with Neutrophils

Figure 6 presents an overview of probable alterations in *P. lutzii* yeast cells interacting with neutrophils. As observed, the figure depicts the integration of metabolic pathways presumably affected by neutrophils in *P. lutzii.* The fungal cells internalized by neutrophils may undergo a metabolic shift where two regulatory enzymes are decreased. The first is 6-phosphofructo-2-kinase (PFK) from glycolysis, and the other is isocitrate dehydrogenase (IDH) from TCA. The enzyme pyruvate dehydrogenase (PDH) exhibited an increase, potentially aimed at supporting the TCA cycle, which might be affected due to the reduced expression of IDH. In addition, malate synthase (MLS) is induced, taking part in anaplerotic reactions that enable cells to use substrates that enter central carbon metabolism; the amino acid synthesis is decreased. Finally, enzymes related to reactive oxygen species protection increased (SOD1, SOD2, SOD5, HSP90, CAT and AAO), suggesting that *P. lutzii* cells may be exposed to elevated levels of reactive species within neutrophils.

### 3.6. Descriptive Analysis of Some Neutrophil Proteins Identified upon Interaction with P. lutzii

We also focused on neutrophil proteins when interacting with *P. lutzii*. Table 4 depicts the proteins that could be identified in neutrophils interacting with *P. lutzii*. Among those proteins, we can see Azurocidin (CAP7_HUMAN), a serine protease with antimicrobial effects. Additionally, defensin 1 (DEF1_HUMAN), a major family of antimicrobial peptides expressed predominantly in neutrophils and epithelial cells, was detected. Additionally, proteins that compose NETs, such as elastase (ELNE_HUMAN), myeloperoxidase (PERM_HUMAN), lactotransferrin (TRFL_HUMAN) and histones, were detected in neutrophils during interaction with *P. lutzii*.

## 4. Discussion

In this work, we focused on the analysis of proteins regulated in *P. lutzii* upon interaction with neutrophils. The environment within neutrophils induces a series of metabolic changes in *P. lutzii*, including proteins related to the response to the formation of oxidative species that leads to metabolic stress. This was also confirmed in *C. albicans* through the use of different approaches [30,31]. To adapt to such conditions, *C. albicans* internalized by neutrophils overexpresses enzymes that detoxify oxidative species, such as SOD [32]. Another study related to the transcriptional response of *Cryptococcus neoformans* in a macrophage infection assay [33] showed that SOD mutant lineages were more sensitive to reactive species in vitro. Lastly, SOD protected *Histoplasma capsulatum* yeast cells from host-derived oxidative stress. Mutant strains lacking the SOD gene reduced the ability of fungi to initiate respiratory infections due to the rapid clearance performed by the immune system [34]. The interaction of *P. brasiliensis* with human PMNs led to an increase in SODs, including SOD1, as seen in the present work. The silencing of SOD1 provoked fungal susceptibility to PMNs, decreased fungal burden in the lung, eliminated fungal burden in the liver and impaired the ability to eliminate intracellular H_2_O_2_ [35]. Similar to our findings, the increase in SOD and other antioxidant proteins was also observed in previous work from our group, as seen in *P. lutzii* during oxidative stress provoked by H_2_O_2_ and macrophage infection [36] and in *P. brasiliensis* during macrophage [37] and murine lung [22] infection. These findings suggest a potential role for those proteins in the defense against oxidative stress found during host infection. Additionally, HSP90 has a pathogenic role in systemic infection since it can interact with serum proteins and change their conformation. This impairs the function of such serum proteins and influences the pathogenicity of *C. albicans* [38]. In addition, HSP90 is present on the cell surface of *Paracoccidioides* spp., and inhibitors reduce fungal load in infected mice [39].

The internalization of *P. lutzii* by neutrophils increased proteins of β-oxidation, as expected. Previous studies have shown that pathogens infecting macrophages can catabolize host lipids [40]. Similarly, previous proteomics analysis from our group revealed a metabolic shift from glycolysis to β-oxidation in *P. brasiliensis* internalized by macrophages [37], as well as in this fungus infecting mouse lungs in vivo [22], both showing an upregulation of enoyl-CoA hydratase. The increase in fatty acid oxidation in our work suggests a possible use of lipids as fuel by *P. lutzii* internalized by neutrophils. Enzymes from the pyruvate dehydrogenase complex increased in *P. lutzii* after neutrophil internalization, similar to the findings in macrophage infection [37]. From this complex, we highlight the enzyme dihydrolipoyl dehydrogenase (DLD), which was previously described as *P. brasiliensis* and *P. restrepiensis* exoantigen [41,42]. Consistent with those findings, DLD was identified among *P. lutzii* secreted proteins [43]. Additionally, DLD was relevant to the phagocytic and microbicidal activities of macrophages [41]. These data suggest that DLD might play a role in host–pathogen interactions.

Our proteomic results depicted an increase in glutamate dehydrogenase (GDH), an enzyme that releases α-ketoglutarate. The latter is critical for maintaining the α-ketoglutarate pool in the cytosol and feeding TCA as an alternative for ATP production in starving pathogenic cells. A high-throughput assay testing the effect of glutamate dehydrogenase deletion in *C. albicans* showed ineffective growth on arginine as a sole carbon and nitrogen source. The enzyme activity favors NADPH production as a way to fulfill the energy requirements of cells under limitations in terms nutrients, such as inside host cells [44]. Additionally, referring to amino acid metabolism, alanine glyoxylate aminotransferase and D-amino acid oxidase (AAO) were increased. The first releases pyruvate, and the latter oxidizes D-amino acids into α-keto acids and ammonia. AAO can avoid the growth inhibitory effect of nutrient limitation and protect pathogen cells from high concentrations of oxidative species in *C. albicans* [45]. Of the proteins discussed so far, GDH and alanine glyoxylate aminotransferase (AGA) were also increased during macrophage infection by *P. brasiliensis* [37]. This suggests that the limited condition of the macrophage and neutrophil environments alters the fungi metabolism similarly.

Two glycolytic enzymes were increased in *P. lutzii* internalized by neutrophils; they are phosphoglycerate kinase (PGK) and glucose-6-phosphate 1-epimerase. PGK can be modulated by stress and under limited nutritional conditions. Higher levels of this enzyme were found in *C. albicans* in conditions of intense need for energy [46]. In addition, PGK can act as an adhesin in bacteria [47] and fungi [48] and is present in the *C. neoformans* [49] and *C. albicans* [50] cell wall. In *P. lutzii*, a secreted PGK was identified as a plasminogen-binding protein [51], suggesting that this protein is a virulence factor, which could account for its increase during the infection of host cells. Conversely, the glycolytic enzyme 6-phosphofructo-2-kinase (PFK) decreased, similarly to the observed in *P. brasiliensis* after macrophage infection [37]; this enzyme is a regulation point of glycolysis, and its repression in the present condition suggests that glycolysis might be damaged and its final product, pyruvate, is depleted.

In this way, the enzyme ATP-citrate lyase (ACL), which sustains the replenishment of the TCA cycle, is increased in *P. lutzii* when internalized by neutrophil. It plays an important role in *C. neoformans* infection, and its absence significantly reduces cell fitness and fungal burden in host cells and makes the pathogen hypersusceptible to fluconazole [52]. Another assay has shown that ACL releases acetyl-CoA in order to feed TCA and is tightly related to the virulence of *C. neoformans* [53]. A similar interpretation can be grasped from the increase in the enzyme 3-isopropylmalate dehydratase, which takes part in the leucine biosynthetic process and may produce energetic intermediaries that run in the TCA as it has aconitase activity. *A. fumigatus* deprived of 3-isopropylmalate dehydratase has its virulence attenuated [54], and *C. neoformans* under the same conditions does not grow in a leucine-limited medium, showing decreased pathogenicity [55].

Another TCA enzyme, succinyl-CoA ligase (SCL), plays a key role in cellular metabolism, and its increase in this assay suggests that the fungal cells internalized by neutrophils possibly undergo severe metabolic changes. This enzyme is overexpressed in several *Paracoccidioides* spp. assays under different conditions, such as during infection of alveolar macrophages [56], oxidative stress [36] and during the process of thermo-differentiation, a condition where the pathogen cells undergo metabolic shift [57,58]. The enzymes from the glyoxylate shunt, malate synthase (MLS) and isocitrate lyase (ICL) increased in *P. lutzii* upon neutrophil internalization. The same was observed in *P. brasiliensis* internalized by macrophages [59] and in *C. albicans* during neutrophil internalization [30]. The role of glyoxylate shunt is to generate gluconeogenic substrates from two-carbon compounds. The shunt is important for phagocytized fungi, serving as an alternative source of carbon, as seen in *S. cerevisiae* and *C. albicans* [40,60]. In *C. neoformans*, ICL increased in fungus isolated from cerebrospinal fluid [61] and after macrophage infection [62]; however, different from *C. albicans* [60], the loss of ICL did not impair fungal virulence [61]. In *P. brasiliensis*, ICL increased during mouse lung infection [22]. The increase in glyoxylate enzymes in our work suggests that *P. lutzii* potentially shifts to a non-glycolytic substrate metabolism to circumvent the lack of carbohydrates within the neutrophils.

The amino acid biosynthesis decreased In *P. lutzii* after neutrophil internalization. The enzymeshorismitee synthase (CS), asparagine synthetase (AS) and glutamine synthase (GS) decreased in the present assay. Fungal cells subjected to stress conditions tend to inhibit the synthesis of amino acids since the process is energetically expensive. Stress conditions mimic the microenvironment faced by the pathogen within host cells during infection. AS decreased in a proteomic approach that assessed metabolic alterations of *P. lutzii* under carbon starvation [63], and GS was also decreased in the same species when cells were treated with an antifungal compound—argentilactone [64]. Both conditions induce stress to the pathogen cells, presumably similar to that encountered inside the immune cells of the host, which is consistent with the decrease in AS in *P. brasiliensis* internalized by macrophages [37].

Another relevant finding is the detection of some proteins in neutrophils after the internalization of *P. lutzii*. One of the identified proteins is Azurocidin, a protein with antimicrobial effects that is rapidly produced by migrating PMN. Additionally, Azurocidin may have a crucial role as a mediator of the initiation of the immune response by alerting the immune system through different mechanisms, such as the activation of monocytes and macrophages [65]. Additionally, proteins present in NET structures were regulated upon neutrophil interaction with *P. lutzii*. Analyses of the role of the protein components of NETs during the internalization of *C. albicans* identified that the proteins elastase, myeloperoxidase, lactotransferrin and histones—also identified in the present work—were involved in the interaction with the fungal cell surface [31]. It has previously been demonstrated that neutrophils induce NET formation upon the internalization of *P. brasiliensis*, presenting a fungistatic effect [17]. ROS generated by NADPH oxidase can trigger the translocation of neutrophil elastase (NE) and myeloperoxidase (MPO) from primary granules to the cytoplasm. MPO and NE together facilitate chromatin decondensation, and chromatin mixes in the cytoplasm with cytoplasmic and granular proteins [66,67]. The presence of these proteins in our work suggests that neutrophil probably triggers NET production to prevent fungal spread.

## 5. Conclusions

In synthesis, we employed mass spectrometry coupled with high-performance liquid chromatography to analyze the response of the dimorphic fungus *P. lutzii* to in vitro interaction with human neutrophils. Quantitative and qualitative changes in the set of proteins related to beta oxidation, stress response, glyoxylate cycle, among others, demonstrated a potential shift in metabolism, as well as the fungus response to the hostile environment of neutrophils and provided new molecules to focus on in new studies involving fungal responses to a host.

## Figures and Tables

**Figure 1 jof-09-01088-f001:**
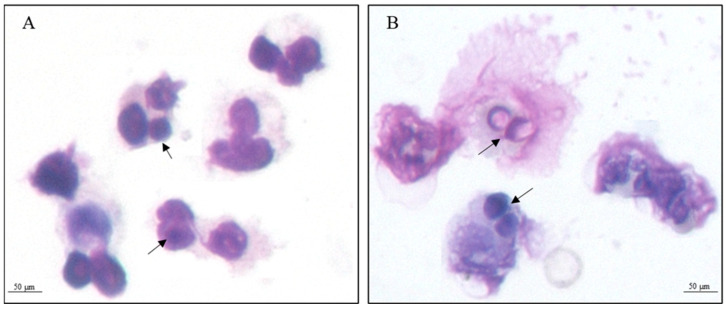
Giemsa-stained smear showing the interaction of *P. lutzii* with neutrophils. (**A**) 2 h and (**B**) 4 h. The images demonstrate activated neutrophils with internalized fungi (black arrows). Insert-100× original magnification.

**Figure 2 jof-09-01088-f002:**
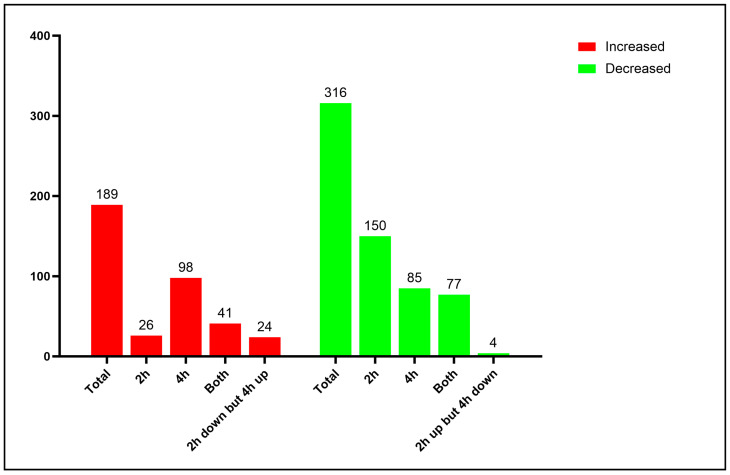
Distribution of increased and decreased proteins identified in proteome. Red: From 189 proteins that increased, 26 and 98 were identified only at 2 h and 4 h of fungal internalization by neutrophil, respectively; 41 proteins were identified at both interaction times and 20 proteins decreased at 2 h but increased at 4 h. Green: From 316 decreased proteins, 150 and 85 were exclusively found at 2 h and 4 h, respectively, of neutrophil internalization; 77 proteins were found at both interaction times and 4 proteins increased at 2 h but decreased at 4 h.

**Figure 3 jof-09-01088-f003:**
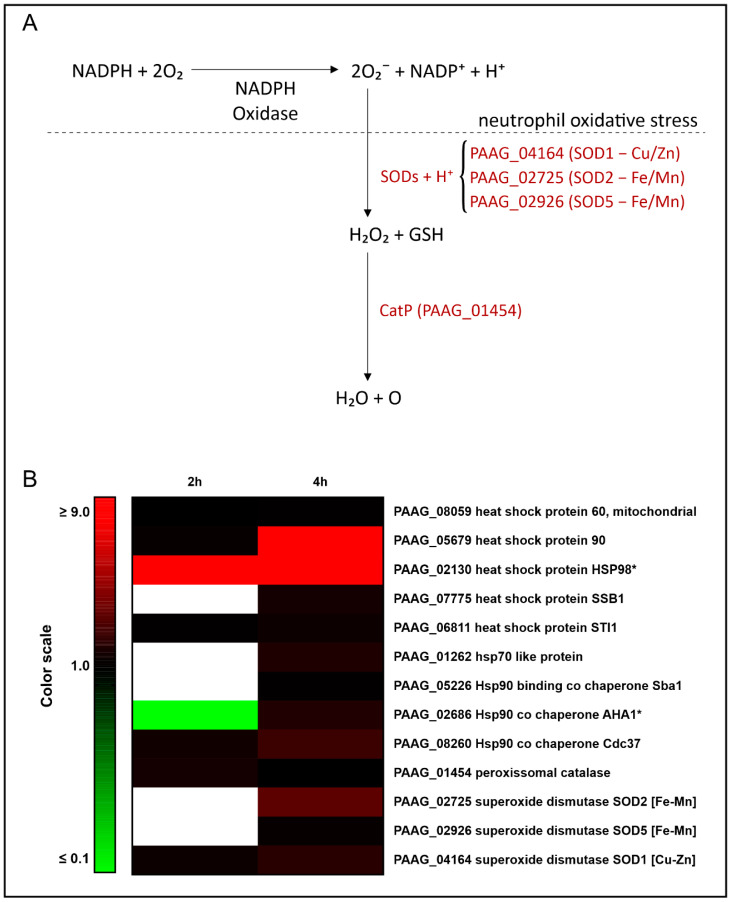
Overview of detoxification mechanisms against oxidative stress in *P. lutzii* during neutrophil internalization. (**A**) This figure summarizes data from proteomic analyses and suggests the mechanism used by this fungus in response to the oxidative stress of PMNs. Superoxide dismutases mediate the detoxification of superoxide into hydrogen peroxide. The degradation of hydrogen peroxide is catalyzed by catalase. (**B**) Heat map of increased stress response and detoxification proteins. Blank cells illustrate proteins not found at 2 h or 4 h. * Indicates protein exclusively found after 2 h and/or 4 h of internalization and not detected in control yeast cells.

**Figure 4 jof-09-01088-f004:**
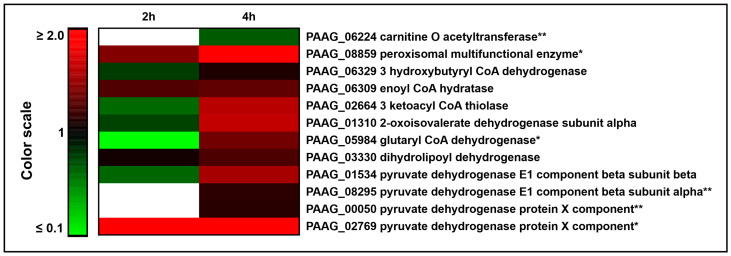
Increase in acetyl-CoA formation in *P. lutzii* during neutrophil internalization. Heat map of increased proteins that are related to acetyl-CoA formation through lipid oxidation and amino acid metabolism. Blank cells illustrate proteins not found at 2 h or 4 h. * Indicates protein exclusively found after 2 h and/or 4 h of internalization and not detected in control yeast cells. ** Indicates proteins that were considered increased since they were not found at 2 h but depicted expression at 4 h of neutrophil interaction.

**Figure 5 jof-09-01088-f005:**
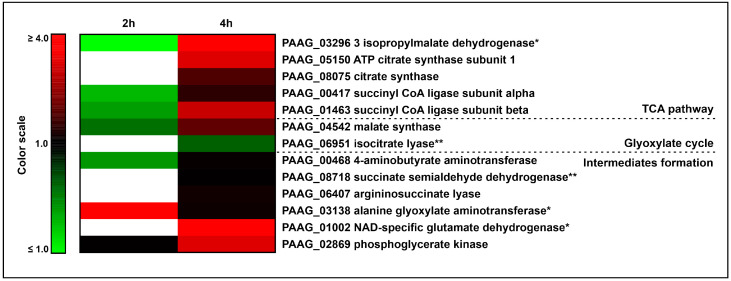
Increase in tricarboxylic acid pathway and glyoxylate cycle and its intermediates formation in *P. lutzii* during neutrophil internalization. Heat map of increased TCA, glyoxylate cycle proteins and those related to intermediates formation. Blank cells illustrate proteins not found at 2 h or 4 h. * Indicates protein exclusively found after 2 h and/or 4 h of internalization and not detected in control yeast cells. ** Indicates proteins that were considered increased since they were not found at 2 h but depicted expression at 4 h of neutrophil interaction.

**Figure 6 jof-09-01088-f006:**
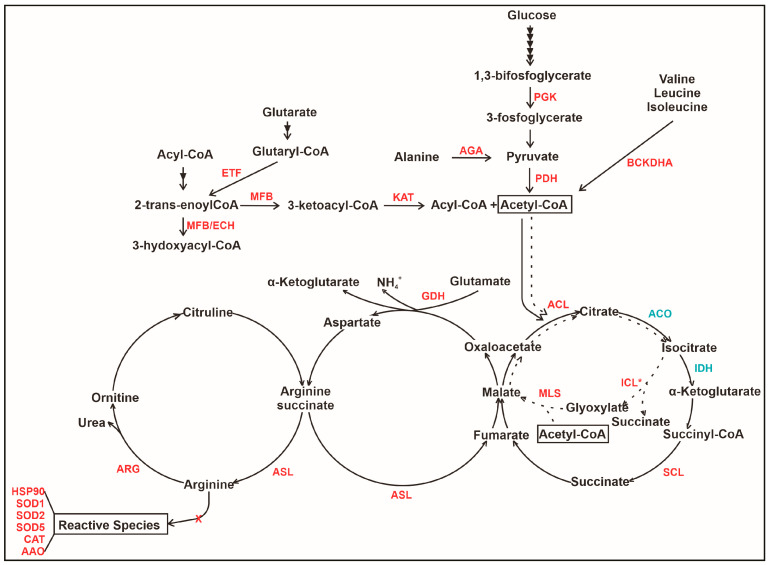
Model depicting presumed metabolic changes during *P. lutzii* yeast cells interacting with neutrophils. The figure highlights proteins that are regulated in metabolic pathways (beta oxidation, TCA, glyoxylate cycle, urea cycle and amino acid metabolism). AAO—D-amino acid oxidase; ACL—ATP citrate lyase; ACO—aconitase; AGA—alanine glyoxylate aminotransferase; ARG—arginase; ASL—arginine succinate lyase; BCKDHA—Oxoisovarelate dehydrogenase; CAT—catalase; ECH—enoyl CoA hydratase; ETF—glutaryl CoA dehydratase; GDH—glutamate dehydrogenase; HSP90—heat shock protein 90; ICL—isocitrate lyase; IDH—isocitrate dehydrogenase; KAT—3 ketoacyl CoA thiolase; MFB—peroxisomal multifuncional enzyme; MLS—malate synthase; PDH—pyruvate dehydrogenase; PGK—phosphoglycerate kinase; SCL—succinyl CoA ligase; SOD1—superoxide dismutase [Cu-Zn]; SOD2—superoxide dismutase SOD2 [Fe-Mn]; SOD5—superoxide dismutase SOD5 [Fe-Mn]. Decreased proteins are blue, and increased proteins are red. * represents proteins that did not appear at 2 h but were regulated at 4 h of neutrophil internalization. The arrow marked with a red X implicates the not conversion of arginine into reactive species due to the increase of ARG.

**Table 1 jof-09-01088-t001:** Increased proteins that are related to stress response and detoxification.

Accession ^a^	Description ^b^	Expression 2 h ^c^	Expression 4 h ^c^
PAAG_08059	heat shock protein 60, mitochondrial	#	1.26
PAAG_05679	heat shock protein 90	1.40	8.76
PAAG_02130	heat shock protein HSP98	*	*
PAAG_07775	heat shock protein SSB1	#	1.84
PAAG_06811	heat shock protein STI1	1.27	1.58
PAAG_01262	hsp70-like protein	#	2.08
PAAG_05226	Hsp90-binding co chaperone Sba	#	1.27
PAAG_02686	Hsp90 co chaperone AHA1	#	2.10
PAAG_08260	Hsp90 co chaperone Cdc37	1.70	2.86
PAAG_01454	peroxisomal catalase	1.79	#
PAAG_02725	superoxide dismutase SOD2 [Fe-Mn]	#	3.78
PAAG_02926	superoxide dismutase SOD5 [Fe-Mn]	#	1.42
PAAG_04164	superoxide dismutase SOD1 [Cu-Zn]	1.57	2.34

^a^ Accession number from FungiDB Database (https://fungidb.org/fungidb/app, accessed on 26 September 2022). ^b^ Protein description according to FungiDB Database (https://fungidb.org/fungidb/app, accessed on 26 September 2022). ^c^ Expression values of differentially expressed *P. lutzii* proteins after 2 h and 4 h of neutrophil contact compared to those that were not incubated with neutrophils. Expression pattern was obtained via ProteinLynx Global Server and normalized with Phosphorylase B. * Proteins exclusively found at 2 h or 4 h of neutrophil interaction. # Proteins not found or not increased at 2 h or 4 h.

**Table 2 jof-09-01088-t002:** Increased proteins related to acetyl-CoA formation.

Accession ^a^	Description ^b^	Expression 2 h ^c^	Expression 4 h ^c^
PAAG_06224	carnitine O acetyltransferase		0.68 **
PAAG_08859	peroxisomal multifunctional enzyme	1.43	*
PAAG_06329	3 hydroxybutyryl CoA dehydrogenase	#	1.12
PAAG_06309	enoyl CoA hydratase	1.27	1.32
PAAG_02664	3 ketoacyl CoA thiolase	#	1.62
PAAG_01310	2-oxoisovalerate dehydrogenase subunit alpha	#	1.65
PAAG_05984	glutaryl CoA dehydrogenase	#	1.38
PAAG_03330	dihydrolipoyl dehydrogenase	#	1.26
PAAG_01534	pyruvate dehydrogenase E1 component subunit beta	#	1.55
PAAG_00050	pyruvate dehydrogenase protein X component	#	1.15 **
PAAG_02769	pyruvate dehydrogenase protein X component	*	*

^a^ Accession number from FungiDB Database (https://fungidb.org/fungidb/app, accessed on 26 September 2022). ^b^ Protein description according to FungiDB Database (https://fungidb.org/fungidb/app, accessed on 26 September 2022). ^c^ Expression values of differentially expressed *P. lutzii* proteins after 2 h and 4 h of neutrophil contact compared to those that were not incubated with neutrophils. Expression pattern was obtained via ProteinLynx Global Server and normalized with Phosphorylase B. * Proteins exclusively found at 2 h or 4 h of neutrophil interaction. # Proteins not found or not increased at 2 h or 4 h. ** Proteins considered increased since they were not found at 2 h but showed expression at 4 h of neutrophil interaction.

**Table 3 jof-09-01088-t003:** Increased proteins related to TCA pathway and its intermediates.

Accession ^a^	Description ^b^	Expression 2 h ^c^	Expression 4 h ^c^
Tricarboxylic-acid pathway			
PAAG_03296	3-isopropylmalate dehydratase	#	*
PAAG_05150	ATP citrate synthase subunit 1	#	3.97
PAAG_08075	citrate synthase	#	2.05
PAAG_00417	succinyl-CoA ligase subunit alpha	#	1.65
PAAG_01463	succinyl CoA ligase subunit beta	#	3.67
Glyoxylate cycle			
PAAG_04542	malate synthase	0.78	2.29
PAAG_06951	isocitrate lyase	#	0.81 **
Intermediates formation			
PAAG_00468	4 aminobutyrate aminotransferase	#	1.21
PAAG_08718	succinate semialdehyde dehydrogenase		1.14 **
PAAG_06407	argininosuccinate lyase	#	1.32
PAAG_03138	alanine glyoxylate aminotransferase	*	1.26
PAAG_01002	NAD specific glutamate dehydrogenase	#	*
PAAG_02869	phosphoglycerate kinase	#	3.97

^a^ Accession number from FungiDB Database (https://fungidb.org/fungidb/app, accessed on 26 September 2022). ^b^ Protein description according to FungiDB Database (https://fungidb.org/fungidb/app, accessed on 26 September 2022). ^c^ Expression values of differentially expressed *P. lutzii* proteins after 2 h and 4 h of neutrophil contact compared to those that were not incubated with neutrophils. Expression pattern was obtained via ProteinLynx Global Server and normalized with Phosphorylase B. * Proteins exclusively found at 2 h or 4 h of neutrophil interaction. # Proteins not found or not induced at 2 h or 4 h. ** Proteins considered increased since they were not found at 2 h but are expressed at 4 h of neutrophil interaction.

**Table 4 jof-09-01088-t004:** Neutrophil proteins that were identified upon interaction with *P. lutzii*.

Acession ^a^	Description ^b^	Amount (Fmol)	Amount (Ngrams)
H4_HUMAN	Histone H4	647.15	7.36
TRFL_HUMAN	Lactotransferrin	404.55	32.39
H2B1B_HUMAN	Histone H2B type 1 B	379.01	5.29
HSP7C_HUMAN	Heat shock cognate 71 kDa protein	328.01	23.33
H2A2C_HUMAN	Histone H2A type 2-C	319.81	4.47
PERM_HUMAN	Myeloperoxidase	234.63	19.91
ELNE_HUMAN	Neutrophil elastase	225.46	6.57
RL40_HUMAN	Ubiquitin 60S ribosomal protein L40	215.02	3.23
SREK1_HUMAN	Splicing regulatory glutamine/lysine-rich protein 1	207.21	12.32
LYSC_HUMAN	Lysozyme C	180.92	3.07
H33_HUMAN	Histone H3.3	144.39	2.22
DEF1_HUMAN	Neutrophil defensin 1	131.28	1.38
H32_HUMAN	Histone H3.2	81.10	1.25
CAP7_HUMAN	Azurocidin	78.46	2.15
SPIN1_HUMAN	Spindlin-1	44.74	1.33
1433Z_HUMAN	14-3-3 protein zeta/delta	39.23	1.10
PNKP_HUMAN	Bifunctional polynucleotide phosphatase/kinase	28.24	1.63
1433B_HUMAN	14-3-3 protein beta/alpha	25.68	0.72
HSP72_HUMAN	Heat shock related 70 kDa protein 2	15.92	1.12
PYGB_HUMAN	Glycogen phosphorylase, brain form	6.72	0.65
NELFE_HUMAN	Negative elongation factor E	4.85	0.21

^a^ Accession number from Uniprot Database (https://www.uniprot.org/, accessed on 28 September 2022). ^b^ Protein description according to Uniprot Database (https://www.uniprot.org/, accessed on 28 September 2022).

## Data Availability

The datasets presented in this study can be found in online repositories. The names of the repository/repositories and accession number(s) can be found below: http://www.peptideatlas.org/ (accessed on 15 February 2023), PASS03809.

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
