# Peer review of "The Response of Paracoccidioides lutzii to the Interaction with Human Neutrophils"

_jof, 2023, doi:10.3390/jof9111088_

Round 1
Reviewer 1 Report
Comments and Suggestions for Authors
The work by Silva et al. describes the proteome analysis of Paracoccidioides lutzii recovered from neutrophils after 2 h and 4 h of coincubation, as compared with yeasts alone cultivated under the same conditions. They also show neutrophil proteins in their analysis. The authors concluded that P. lutzii yeast cells living within neutrophils for a few hours suffered a shift in metabolism from glycolysis to beta oxidation, as they observed increased amounts of enzymes characteristic of the glyoxylate cycle, and also related to detoxification of oxidative and heat shock stress. The data can contribute nicely to the knowledge of the protein expression response of P. lutzii phagocytosed by neutrophils, especially if specific points are clarified and/or modified, as listed below.
1. How virulent is P. lutzii (ATCC MYA-826) in mice? How is it kept medium- and long-term in the lab? Please add extra information about in methodology. 2. l. 106: Was RPMI supplemented with FBS? In l.104, please specify the serum used to opsonize the yeast cells. 3. In "preparation of protein extract" (l.108) or in Results (Figure 1), please explain how (or if) bound yeasts that have not been phagocytosed (not mentioned or shown in Figure 1, however likely to exist) were washed out. If they are part of the analyzed yeasts, be specific about it. 4. You use high (but acceptable) FDR rates of 4% and a very low cut off for differentially expressed proteins of 1,2%. Conclusions about the proteins related to the acetyl Con A metabolism have been taken from enzymes hardly overexpressed using that cut off. Maybe the authors should increase their parameters or at least soften their conclusions. In adittion, conclusions should be softened throughout the whole paper, even when differences were greater, considering enzymatic assays have not been performed. 5. Is it correct to say that P. lutzii was "infecting" neutrophils (l. 361, for e.g.)? 6. item 3.6. needs to be clarified. There is no mention in the methodology of how the cells were prepared for analysis. Or have these proteins been detected in the P. lutzii proteome? 7. The amount of neutrophil proteins is described in ngrams (Table 4), but the amount of P. lutzii proteins in the supplemental tables is described in "score". Can you specify "score"?
English is readable, but can be greatly improved for grammar and style.
Reviewer 2 Report
Comments and Suggestions for Authors
In the manuscript ‘The response of Paracoccidioides lutzii to the interaction with human neutrophils’, the authors analyse proteomic profiles of P. lutzii in confrontation in vitro with human neutrophils. Proteomic profiling of P. lutzii in contact to neutrophils revealed that the yeast cells underwent a shift in metabolism from glycolysis to beta-oxidation, including the glyoxylate shunt. They also upregulates enzymes involved in detoxification of ROS and these, preventing heat shock stress.
In my opinion, presented work is interesting and it is important to the other fungal researchers, working on Paracoccidioides sp. in interaction with human lungs. The carefully thought-out experiments gave the interesting results and the high-throughput proteomic analysis done here is impressive. However, I see few minor points concerning this work, which makes it unclear, as listed below:
The main remark: the authors suggest that neutrophils are the first cells in contact with P. lutzii, next replaced by macrophages (row 38). Meanwhile, in their previous work from 2019y. (PMID: 30804901), the authors claim that macrophages act for the beginning. Please, explain this contradiction.
In the section Discussion, hardly find any comparison with Your previous results, such as proteome of P. brasiliensis upon interaction with macrophages, to show the similarities and differences.
In proteins detected in neutrophils there is no inflammatory response markers, such as cytokines, what is surprising.
Minor points:
Row 67: produced evidences – sounds awkward
Row 104: which kind of serum was used? The content of proper antibodies in serum is crucial for opsonisation and efficient recognition by host
Row 420: The infection of...increased the production of enzymes involved in beta-oxidation (of course) in P. lutzii cells. Please, rewrite this shortcut.
Row 480-1:...important for...pathogens... S. cerevisiae and C. albicans – S. cerevisiae is considered as a non-pathogenic yeast. Please, check citation order (40,60) and it is really related to the proper citation.
Row 503: initiation of the immune response (citation needed)
Comments on the Quality of English LanguageThe English is quite good to read and understand. Few minor points I have addressed in the review.
Reviewer 3 Report
Comments and Suggestions for Authors
The manuscript entitled “The response of Paracoccidioides lutzii to the interaction with human neutrophils" written by Souza Silva et al.
The manuscript is descriptive, describing protein level changes in P. lutzii after neutrophil infection, but none of those proteins or biochemical pathways were evaluated to see whether or not their biological relevance in the infection process Without this information, it is difficult to interpret the results.
Figures lack formal statistical analysis; the authors need to do this analysis to determine whether or not the differences were statistically significant.
In the text, it is written in line 240, “example, increased more than 8 times in yeast cells upon 6 hours," but Table 1 only describes 2 and 4 h. I consider that values in the tables with 2 decimals are more than enough, and those values in the table 1 (and also for the other tables) lack of standard error/desviation. Is difficult to interpret the data without statistical analysis.
Moreover, “3.2. Interaction with human neutrophils promotes increase of transcripts of the oxidative stress response in P. lutzii “ but the authors only showed protein, don´t they?
How can the author experimentally test the role of the antioxidant systems induced in P. lutizii during the neutrophil infection? The authors can incubate P. lutzi for 30 minutes with N-acetilcisteine or another antioxidant and thereafter assay neutrophils to determine the survival of the fungus.
Lane 384. Avoid contractions “neutrophil’s proteins.".
Round 2
Reviewer 1 Report
Comments and Suggestions for Authors
The manuscript has been nicely edited.
Comments on the Quality of English LanguageMy concerns have been answered and the ms has been edited accordingly.
Reviewer 3 Report
Comments and Suggestions for Authors
The manuscrit can be accepted in the present form